# Frequency Decomposition and Enhancement for Time Series Generation Using Diffusion Models

## Abstract

Time series data are essential in domains such as finance, healthcare, energy management, climate prediction, and AIOps, yet the scarcity of large-scale, high-quality training datasets often restricts the performance of machine learning solutions. Synthetic data generation, particularly through diffusion models, has become a promising strategy to address these limitations. Diffusion-based models have showcased impressive results, but face challenges in capturing diverse frequency components and retaining high-frequency details during noise accumulation. To address these issues, we propose a multi-stage diffusion framework named Frequency Decomposed and Enhanced Diffusion (FDEDiff), which explicitly decomposes time series into low- and high-frequency signals and emphasizes preserving fine-grained temporal patterns. Our method first trains an unconditional generator on coarse, periodic low-frequency signals, then incorporates an enhancement mechanism to synthesize precise high-frequency details. This two-stage approach systematically handles complex temporal variations, allowing FDEDiff to produce more accurate, realistic, and diverse time series. We conduct extensive experiments on publicly available real-world datasets, demonstrating that FDEDiff not only outperforms state-of-the-art generative methods in various evolution metrics but also exhibits superior adaptability across different time series domains. An ablation study confirms the effectiveness of frequency decomposition and high-frequency enhancement, underscoring the advantage of exploiting multi-resolution insights. Our findings expand the application scope of diffusion models for time series generation tasks, offering a flexible solution for data augmentation under privacy and sensitivity constraints. We have made our code anonymously available at https://github.com/FDEDiffCode/FDEDiff.

## 1 Introduction

Time series data is critical in many applications across various domains such as finance, healthcare, energy management, climate prediction, and AIOps (Yan et al. (2024); Zheng et al. (2024)). With the rapid advancement of artificial intelligence, machine learning methods have become increasingly prevalent for analyzing and interpreting time series (Wang et al. (2024); Zhang et al. (2024)). However, developing robust machine learning solutions becomes challenging when large-scale, high-quality training datasets are unavailable due to privacy constraints or data sensitivity. To address this limitation, synthetic time series generation emerges as a viable approach and draws significant attention in recent research (Ang et al. (2023)). Most works are built on top of classical generative frameworks such as VAE and GAN.

Besides VAE and GAN, the diffusion model has recently emerged as a powerful and increasingly popular class of generative methods. The diffusion model has demonstrated remarkable performance across a wide range of domains, including image (Dhariwal & Nichol (2021)) and video (Harvey et al. (2022)) generation. Motivated by their success, recent studies have explored the application of the diffusion model to time series generation tasks. Diffusion-based approaches can outperform other generative models, offering higher generation quality. For example, Diffusion-TS (Yuan & Qiao (2024)) replaces the standard predictor in the diffusion model with a multi-layer architecture consisting of seasonal and trend component predictors; it generates the final time series by summing the components from each layer. Diff-MTS (Ren et al. (2024)) introduces a conditional diffusion framework to synthesize time series based on the health state sequence of each device. MR-Diff (Shen et al. (2024)) employs a multi-resolution diffusion strategy, progressively generating target segments from coarse to fine scale to capture smooth trends and detailed features.

While existing diffusion-based models incorporate architectures such as Transformers, convolutional networks, and RNNs to better capture temporal patterns, two key challenges still hinder their ability to generate high-quality time series. First, it remains difficult for diffusion models to unconditionally generate time series that exhibit diverse seasonal and trend components, along with irregular residual variations. Second, due to the progressive noise accumulation in

the diffusion process, high-frequency components in time series often vanish early during forward diffusion (Galib et al. (2024)). This results in overly smoothed outputs that lack local detail and compromise the diversity of the generated data. Addressing these limitations is essential for advancing the fidelity and utility of diffusion-based time series generation.

To solve the challenges mentioned above, in this paper, we propose a two-stage diffusion model framework named **F**requency **D**ecomposed and **E**nhanced **Diff**usion model (**FDEDiff**). FDEDiffexplicitly decomposes time series into low-frequency and high-frequency parts and employs a high-frequency enhancement mechanism to preserve fine-grained temporal details. These two components are modeled separately through a two-stage structure. Acknowledging the tendency of the diffusion process to generate overly smooth outputs, FDEDifffirst trains an unconditional generator on coarse, periodic low-frequency signals. In the second stage, high-frequency-enhanced sequences are synthesized with guidance from the previously generated low-frequency components. This coarse-to-fine strategy enables the model to generate more accurate and realistic time series.

Our contributions are summarized as follows: (i) We propose FDEDiff, the first framework to integrate frequency-domain decomposition with a multi-stage diffusion model pipeline for time series generation. (ii) By sequentially generating data with different numbers of frequency components, FDEDiffcan synthesize time series with diverse seasonal and trend characteristics. (iii) Extensive experiments on public real-world datasets demonstrate that FDEDiffoutperforms state-of-the-art time series generative models. Furthermore, our ablation study shows the effectiveness of the frequency decomposition and enhancement strategy.

## 2 RELATED WORK

### 2.1 TIME-SERIES GENERATION

Time series data are ubiquitous across various domains in the real world. However, many time series tasks face the challenge of data scarcity due to issues such as data privacy and high acquisition costs(Alaa et al., 2021). To address these limitations and enable robust machine learning solutions when large-scale, high-quality training datasets are unavailable, synthetic time series generation has emerged as a vital research area Ang et al. (2023).To capture the highly complex features and dependencies inherent in input time series, existing works commonly employ self-supervised or unsupervised deep learning models for data generation.

Among these, Generative Adversarial Networks (GANs) have been widely applied to both image and time-series generation tasks Wu et al. (2018); Brock et al. (2018); Esteban et al. (2017). Early contributions, such as that by Morgren et al. Mogren (2016), combined Recurrent Neural Networks (RNNs) with GANs to synthesize sequential data like music. Specifically for time series, TimeGAN Yoon et al. (2019) utilizes GANs to effectively capture the unique temporal correlations present in time-series data through an adversarial training framework.

Meanwhile, Variational Autoencoders (VAEs) also constitute a significant class of generative models extensively used in various generative tasks Li et al. (2023); Desai et al. (2021); Lee et al. (2023). For instance, TimeVAE Desai et al. (2021) achieved notable generation results by designing an interpretable time-series structure, demonstrating the efficacy of VAE-based approaches in modeling complex temporal dynamics. These classical generative frameworks have laid the foundation for synthetic time series data creation.

### 2.2 DIFFUSION MODELS

Denoising Diffusion Probabilistic Models (DDPMs) have gained significant traction following their success in image generation, prompting their adaptation to time-series tasks (Lin et al., 2024; Ren et al., 2024; Shen et al., 2024). The diffusion mechanism excels at modeling intricate dependencies, making it well-suited for generating high-fidelity time series.

For example, DiffSTG (Wen et al., 2023) applies diffusion models to spatio-temporal forecasting, capturing both spatial and temporal dynamics. Similarly, CSDI (Tashiro et al., 2021) leverages diffusion for time-series imputation, enhancing the robustness of missing value predictions. In the realm of time-series generation, DiffusionTS (Yuan & Qiao, 2024) proposes a non-autoregressive Transformer-based diffusion model, incorporating a Fourier-based loss function to refine the reconstruction of temporal patterns during the denoising process. Diff-MTS Ren et al. (2024) proposed a conditional diffusion framework tailored based on specific health state sequences of devices, showcasing the model's ability to generate data conditioned on external factors. Furthermore, (Galib et al., 2024) tackled the issue of high-frequency component dissipation in time-series generation by introducing a frequency-enhanced diffusion strategy to preserve fine-grained details. These advancements highlight the versatility of diffusion models in addressing

diverse challenges in time-series synthesis, offering superior generation quality compared to traditional generative frameworks.

## 3 PRELIMINARY

### 3.1 PROBLEM STATEMENT

Let $\mathbf{X} = [\mathbf{x}_0, \mathbf{x}_1, \ldots, \mathbf{x}_{W-1}] \in \mathbb{R}^{W \times M}$ denote a multivariate time series instance with length $W$ and dimension $M$, where $\mathbf{x}_j$ represents the $M$-dimensional data points at time step $j$. Given a training dataset containing $n$ instances, $\mathbf{D} = \{\mathbf{X}^i\}_{i=1}^n$, our target is to train a diffusion-based generator that maps Gaussian noise vectors to the instances similar to those in $\mathbf{D}$.

### 3.2 DIFFUSION FRAMEWORK

**Denoising Diffusion Probabilistic Models.** The classical diffusion model, known as the Denoising Diffusion Probabilistic Model (DDPM, Ho et al. (2020)), is a latent-variable model consisting of forward diffusion and backward denoising processes. In the forward process, DDPM progressively transforms an input sample $x^0$ into a Gaussian noise vector $x^T$ through $T$ steps of sampling:

$$q(x^t|x^{t-1}) = \mathcal{N}(x^t; \sqrt{1-\beta_t}\, x^{t-1}, \beta_t\, \mathbf{I}), t = 1, 2, \cdots, T. \tag{1}$$

$\beta_t$ is a predefined variance schedule. The forward process equation can be rewritten as $q(x^t|x^0) = \mathcal{N}(x^t; \sqrt{\bar{\alpha}_t}\, x^0, (1-\bar{\alpha}_t)\, \mathbf{I})$, where $\bar{\alpha}_t = \prod_{s=1}^t (1 - \beta_s)$. We can therefore simply obtain $x^t$ by applying the re-parameterization trick:

$$x^t = \sqrt{\bar{\alpha}_t}\, x^0 + \sqrt{1-\bar{\alpha}_t}\, \epsilon, \epsilon \sim \mathcal{N}(0, \mathbf{I}). \tag{2}$$

The backward process starts from a Gaussian noise vector $x^T$ and generates the data sample $x^0$ by $T$ denoising steps. $x^{t-1}$ is sampled from the following normal distribution:

$$p_\theta(x^{t-1}|x^t) = \mathcal{N}(x^{t-1}; \mu_\theta(x^t, t), \sigma_t^2\, \mathbf{I}), t = 1, 2, \cdots, T. \tag{3}$$

Here, the neural network $\mu_\theta(x^t, t)$ is the mean of the normal distribution, and the variance is fixed as $\sigma_t^2\, \mathbf{I}$, where $\sigma_t^2 = \frac{1-\bar{\alpha}_{t-1}}{1-\bar{\alpha}_t}\beta_t$. Time series generation models usually use a $x^0$ predictor (Benny & Wolf (2022); Shen et al. (2024); Yuan & Qiao (2024)) to compute the mean value. Specifically, a neural network $x_\theta(x^t, t)$ is trained to estimate the input sample $x^0$, and then the mean of the normal distribution for sampling is obtained:

$$\mu_\theta(x^t, t) = \frac{\sqrt{1-\beta_t}\,(1-\bar{\alpha}_{t-1})}{1-\bar{\alpha}_t}\, x^t + \frac{\beta_t\, \sqrt{\bar{\alpha}_{t-1}}}{1-\bar{\alpha}_t}\, x_\theta(x^t, t). \tag{4}$$

The parameter $\theta$ is learned by minimizing the following loss:

$$\mathcal{L} = \mathbb{E}_{t,x^0,\epsilon}\left[||x^0 - x_\theta(x^t, t)||^2\right]. \tag{5}$$

**Conditional Diffusion Models.** Conditional diffusion models generate new time series based on existing observations as conditions, *e.g.*, imputation (Tashiro et al. (2021)), forecasting (Shen et al. (2024)), and indicator-conditional generation (Ren et al. (2024)). In conditional diffusion models, the backward process where the condition $c$ serves as the posterior is as given:

$$p_\theta(x^{t-1}|x^t, c) = \mathcal{N}(x^{t-1}; \mu_\theta(x^t, t|c), \sigma_t^2\, \mathbf{I}). \tag{6}$$

A straightforward implementation of the conditional predictor involves fusing the embedding of $c$ into the predictor of the unconditional diffusion model, which jointly encodes the predicted value.

## 4 FDEDIFF: FREQUENCY DECOMPOSED AND ENHANCED DIFFUSION MODEL

In this section, we introduce FDEDiff, a novel data generation framework that integrates diffusion models with frequency-domain processing for time series. To enhance the generation quality of diffusion models for time series, FDEDiff decomposes each sample $\mathbf{X}$ into the low-frequency part $\mathbf{X}_L$ and the augmented high-frequency amplified part $\mathbf{X}_a$, and these two components are modeled using two cascaded diffusion models, as illustrated in Figure 1. The low-frequency model (upper part of Figure 1) is an unconditional diffusion model that captures the smooth and seasonal characteristics of the time series sample, while the high-frequency model (lower part of Figure 1) is a conditional diffusion model focusing on fine-grained details.

Figure 1: Proposed FDEDiffpipeline for generating time series

## 4.1 FREQUENCY DOMAIN DECOMPOSITION AND ENHANCEMENT

For the given time series sample $\mathbf{X} \in \mathbb{R}^{W \times M}$, where $W$ is the length and $M$ is the feature dimension. We first apply the Fourier transform to convert the time-domain signal into the frequency domain for each dimension $m$:

$$\mathcal{F}_m = \text{FFT}(\mathbf{X}_m) \tag{7}$$

where $\mathcal{F}_m \in \mathbb{C}^W$ is the frequency-domain representation of the $m$-th feature. It reveals the amplitude and phase of periodic patterns at $M$ frequencies.

We next define the low-frequency part as the first $k$ frequency components of the spectrum, where $k = \lfloor \alpha W \rfloor$ (the low-frequency ratio $\alpha$ is a hyperparameter). The remaining $W - k$ components constitute the high-frequency part. By setting all high-frequency components to 0, we can obtain the low-frequency representation $\mathcal{F}_L$ and the low-frequenct part of the sample $\mathbf{X}_L$ can be computed by the inverse Fourier transform:

$$\mathcal{F}_{L,m} = \left[ \hat{\mathcal{F}}_d[0:k], 0 \right] \tag{8}$$

$$\mathbf{X}_{L,m} = \text{IFFT}(\mathcal{F}_{L,m}) \tag{9}$$

The high-frequency component $\mathbf{X}_H$ can be obtained using a similar procedure as for $\mathbf{X}_L$. Inspired by FIDE (Galib et al. (2024)), FDEDiffamplifies the high-frequency part in a time series segment to preserve more fine-grained information during the diffusion process. Specifically, $\mathbf{X}_H$ is scaled by a coefficient $\lambda$ ($\lambda$ is a hyperparameter greater than 1) and then added to $\mathbf{X}_L$ to obtain the high-frequency-amplified signal $\mathbf{X}_a$:

$$\mathbf{X}_a = \lambda \mathbf{X}_H + \mathbf{X}_L \tag{10}$$

FDEDifffirst trains a diffusion model on the low-frequency dataset $\mathbf{D}_L = \{\mathbf{X}_L^i\}_{i=1}^n$ (§ 4.2). Then, guided by the low-frequency signals, FDEDifftrains a conditional diffusion model on the high-frequency-amplified dataset $\mathbf{D}_a = \{\mathbf{X}_a^i\}_{i=1}^n$ to generate the final time series segments (§ 4.3, § 4.4).

## 4.2 LOW-FREQUENCY TIME SERIES $\mathcal{Y}_L$ GENERATION

As shown in the upper part of Figure 1, this module aims to generate the low-frequency component of the time series data. First, for the extracted low-frequency time series $\mathbf{X}_L$, we draw inspiration from the diffusion process described in Section 3.2. Specifically, we progressively add Gaussian noise $\boldsymbol{\epsilon} \sim \mathcal{N}(\mathbf{0}, \mathbf{I})$ to $\mathbf{X}_L$, obtaining the noisy low-frequency sequence $\mathbf{X}_{L,t}$ after $t$ steps of the forward diffusion process.

To effectively model the dependencies between different variables in the multivariate time series and the dynamic evolution within the time series, we design a Transformer-based network structure as the $x_0$ predictor. This network takes $\mathbf{X}_{L,t}$ and the current diffusion step $t$ as input. Initially, we employ a Residual Block (ResBlock) to fuse and encode $\mathbf{X}_{L,t}$ and the time step $t$, resulting in an embedding representation that simultaneously contains sequence information and temporal information.

Subsequently, the encoded embedding information is fed into a stacked structure containing $N$ Transformer encoder layers. Each encoder layer includes a Self-Attention module(Vaswani et al. (2017)) to capture the interactions between different time steps and different variables within the low-frequency component. Following the Self-Attention module, we apply the Add & Norm operation for residual connection and layer normalization to capture time information. Inspired by AdaLayer(von Platen et al. (2022)), we utilize AdaLayerNorm for more refined feature normalization and adjustment. Finally, after a feed-forward network, we obtain the output of each encoder layer.

After processing through $N$ encoder layers, we obtain the final output $\mathbf{Y}_L$, which is used as the predicted original low-frequency component $\mathbf{X}_L$ during the low-frequency training phase. It is important to emphasize that we independently train the low-frequency component generation module during the model training phase. Taking the real low-frequency component $\mathbf{X}_L$ as input, the training objective is to make the model's prediction $\mathbf{Y}_L$ as close as possible to $\mathbf{X}_L$, thereby learning the distribution of the low-frequency time series. Formally, this process can be expressed as a conditional denoising diffusion probabilistic model (DDPM) applied to the low-frequency component:

$$\mathbf{Y}_L = \text{DDPM}_{\text{LF}}(\mathbf{X}_L) \tag{11}$$

where $\text{DDPM}_{\text{LF}}$ denotes the low-frequency diffusion model responsible for capturing the underlying distribution and temporal dependencies of the low-frequency signal.

## 4.3 HIGH-FREQUENCY-AMPLIFIED TIME SERIES $\mathcal{Y}_H$ GENERATION

The lower part of Figure 1 depicts the generation process of high-frequency components in temporal data. This high-frequency generation module is designed to capture fine-grained details and rapid fluctuations within the time series. Analogous to the low-frequency generation module, it adopts a diffusion model-based generative framework. However, to effectively integrate the macro-level patterns encoded in the low-frequency components during the generation of high-frequency details, we incorporate a cross-attention mechanism.

Specifically, the diffusion model for the high-frequency component also utilizes a Transformer architecture as the $\mathbf{x}_0$ predictor. Distinct from the low-frequency module, each Transformer decoder layer in the high-frequency module embeds a cross-attention mechanism. During training, this module takes the ground-truth low-frequency component $\mathbf{X}_L$, and during inference, it conditions on the generated low-frequency component $\mathbf{Y}_L$, serving as Key and Value in the cross-attention operation. The intermediate representations within the high-frequency diffusion process act as the Query, thus enabling explicit conditioning on low-frequency information. This cross-attention integration allows the model to synthesize high-frequency details coherent with the overall trends represented by the low-frequency components, ensuring the generated time series maintains consistency across macro and micro temporal scales.

During training, the ground truth of low-frequency component $\mathbf{X}_L$ is used to guide the generation of high-frequency components, enabling the model to learn the distribution of high-frequency details conditioned on varying low-frequency trends. During inference, the generated low-frequency component $\mathbf{X}_{LF}$ from the low-frequency module serves as the condition to achieve end-to-end temporal data generation.

$$\mathbf{Y}_a = \begin{cases} \text{DDPM}_{\text{HF}}(\mathbf{X}_a, \mathbf{X}_L), & \text{Train} \\ \text{DDPM}_{\text{HF}}(\mathbf{X}_a, \mathbf{Y}_L), & \text{Infer} \end{cases} \tag{12}$$

Here, $\text{DDPM}_{\text{HF}}$ represents the high-frequency denoising diffusion model, and $\mathbf{X}_{LF}$ is the low-frequency component that serves as a condition to guide the generation of the high-frequency component. During this process, the generated high-frequency component incorporates both the original high-frequency information and the influence of the low-frequency component, ensuring overall consistency in the generated sequence. At each step, the model updates the

high-frequency component based on the current values of both the high-frequency and low-frequency components, ensuring the accurate generation of fine details.

### 4.4 FINAL TIME SERIES $\mathcal{Y}$ GENERATION

After model training, FDEDiffperforms data generation during inference in two sequential stages. First, the low-frequency diffusion model generates the low-frequency component $\mathbf{Y}_L$. Since the ground-truth low-frequency component $\mathbf{X}_L$ is unavailable, the cross-attention module within the high-frequency model uses the generated low-frequency data $\mathbf{Y}_L$ from the low-frequency model as a conditioning signal to generate the final enhanced high-frequency component $\mathbf{Y}_a$.

It is important to note that the high-frequency model produces an enhanced high-frequency component $\mathbf{Y}_a$. The final reconstructed time series $\mathbf{Y}$ is recovered using the parameter $\lambda$, formulated as follows:

$$\mathbf{Y} = \frac{\mathbf{Y}_a - \mathbf{Y}_L}{\lambda} + \mathbf{Y}_L \tag{13}$$

### 4.5 LIMITATIONS

FDEDiffhas several notable limitations: First, it can only generate fixed-length segments matching the training data and does not support arbitrary-length generation. Second, due to the lack of explicit modeling for dependencies among multiple variables, FDEDiffperforms poorly on high-dimensional generation tasks. Finally, FDEDiffdoes not support conditional generation, making it difficult to control the type or characteristics of the generated time series.

## 5 EXPERIMENTAL EVALUATION

In this section, to evaluate the quality of the time series data generated from FDEDiff, we conduct comprehensive experiments on four public real-world datasets. First, we compare the generation quality of FDEDiffwith the state-of-the-art time series generation models in three categories (§ 5.2). Second, we study how the core designs and key parameters in FDEDiffcontribute to its overall performance (§ 5.3).

### 5.1 EXPERIMENT SETUP

**Datasets.** We evaluate all time series generation methods on four real-world datasets collected from diverse domains (Yuan & Qiao (2024); Zhou et al. (2021); Wu et al. (2021)): *ETTh*[1] contains 2 years of hourly readings from electricity transformers, including 6 external load features and oil temperature, for a total of 7 dimensions. *Stocks*[2] consists of daily Google stock-price records from 2004 to 2019, providing 6 feature dimensions per record. *fMRI*[3] is a standard causal-discovery benchmark; we select a simulated blood-oxygen-level-dependent scenario from the original dataset that has 50 dimensions. *Electricity*[4] records the hourly power consumption of 321 customers (dimensions) for 3 years.

We apply PCA to the *Electricity* dataset and extract the 30 most informative components as a reduced feature set that keeps model training within a reasonable computational budget.

**Baselines.** In the experiments, we select three representative state-of-the-art baselines in different categories. All of them are unconditional time series generation models: Diffusion-TS (Yuan & Qiao (2024)) employs a Transformer as the $x_0$ predictor in the diffusion process. It improves the decoder to produce seasonal and trend components for the generated time series data. TimeVAE (Desai et al. (2021)) implements multiple decoders in VAE to transform a latent variable into seasonal, trend, and residual components of the time series sequence. TimeGAN (Yoon et al. (2019)) jointly trains an AE and a GAN. The AE learns a mapping from the original time series data to a temporal latent space, while the GAN is trained to generate latent variable samples.

**Performance Metrics.** To evaluate the quality of the generated data, we use the following three metrics to measure the discrepancy between the synthetic and the original dataset: Context-Fréchet Inception Distance (**C-FID**, Jeha et al. (2022)) score computes the statistical difference between embeddings of time series that fit into the local context. Auto Correlation Difference (**ACD**, Lai et al. (2018)) computes autocorrelation matrices for the generated and the original

---

[1] https://github.com/zhouhaoyi/ETDataset/blob/main/ETT-small/ETTh1.csv
[2] https://finance.yahoo.com/quote/GOOG/history/?p=GOOG
[3] https://www.fmrib.ox.ac.uk/datasets/netsim/
[4] https://github.com/thuml/Autoformer

time series over multiple lags, and then aggregates their differences to quantify per-dimension periodic similarity. Cross Correlation Difference (**CCD**, Yuan & Qiao (2024)) measures how well the generated time series maintain inter-dimensional dependencies by computing the difference between the cross correlation matrices of two datasets.

**Configurations.** We train all models on a single NVIDIA A800 GPU (80GB). We adopt the parameter settings recommended in public repositories for all baselines for each dataset. Because both FDEDiffand Diffusion-TS employ the Transformer structure as the $x_0$ predictor in the diffusion process, we align their model sizes for fairness (Table 1). In FDEDiff, the number of low-frequency components is set to $\alpha = 2\%$ of the window length, and the high-frequency enhancement factor is fixed to $\lambda = 2$. We employ Adam as the optimizer in FDEDiff. The learning rate is scheduled using a combination of linear warmup and the ReduceLROnPlateau strategy.

Table 1: Model Parameters in FDEDiffand Diffusion-TS.

| Model | Hidden Size | Attention Heads | Layers | Diffusion Steps |
|---|---|---|---|---|
| FDEDiff | 128 | 8 | 4 | 500 |
| Diffusion-TS | 96 | 4 | 4 + 3 | 1000 |

## 5.2 UNCONDITIONAL TIME SERIES GENERATION EVALUATION

This part evaluates the data generation quality of FDEDiffagainst the baseline methods. Specifically, each dataset is first segmented by sliding windows of different lengths (24, 64, 96, 192, and 336). We then report the performance (C-FID, ACD, and CCD) for each method under these varied window lengths. Because computing C-FID involves training a representation-learning model (Jeha et al. (2022)), we calculate it five times for each experimental setting and report the average score. The results with $95\%$ confidence intervals as the error bars are shown in the § A.1 (Table 3).

table 2 presents the evaluation results of all methods across different datasets and window lengths. Compared to the other two metrics (ACD and CCD), C-FID better reflects the overall quality of generated data; thus, we primarily use C-FID as the primary evaluation criterion. FDEDiffgenerates higher-quality data than baseline methods across almost all experiment settings. This superior performance can be attributed to two key factors: First, the Transformer-based predictor in FDEDiffeffectively models long sequences using self-attention and cross-attention mechanisms. This is confirmed by the results showing that Diffusion-TS, which also employs a Transformer predictor, generally outperforms VAE-based and GAN-based methods. Second, the frequency-domain decomposition and enhancement mechanism in FDEDiffeffectively guides generation by leveraging low-frequency information while preserving more high-frequency details. § 5.3 further investigates the contribution of this mechanism to overall model performance.

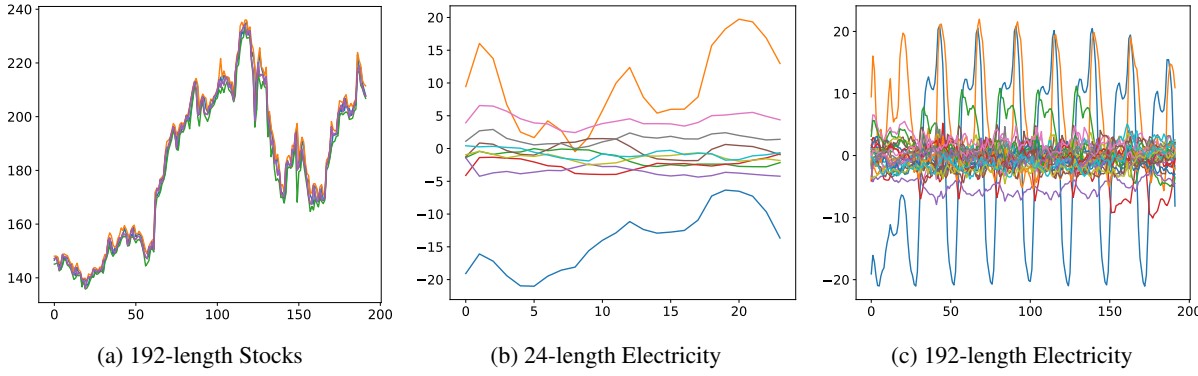

(a) 192-length Stocks     (b) 24-length Electricity     (c) 192-length Electricity

Figure 2: Samples in the Stocks and the Electricity Datasets.

When the window length is 24, the performance of FDEDiffis only comparable to, or even worse than, the baselines, as seen on the Stocks and the Electricity datasets. Figure 2b shows that such a short window exhibits no apparent periodicity and provides few frequency-domain components. As a result, the low-frequency part cannot represent the smooth structure of the window, and the frequency-domain-based optimizations in FDEDiffare ineffective in these cases.

Table 2: Results on Multiple Datasets and Window Lengths (**Lower** metric indicates better performance).

| Dataset | Length | FDEDiff | | | Diffusion-TS | | | TimeVAE[a] | | | TimeGAN | | |
|---|---|---|---|---|---|---|---|---|---|---|---|---|---|
| | | C-FID | ACD | CCD | C-FID | ACD | CCD | C-FID | ACD | CCD | C-FID | ACD | CCD |
| ETTh | 24 | **0.085** | **0.015** | **0.030** | 0.138 | 0.015 | 0.050 | 1.697 | 0.132 | 0.119 | 0.799 | 0.033 | 0.290 |
| | 64 | **0.131** | **0.010** | **0.032** | 0.248 | 0.014 | 0.058 | 1.124 | 0.122 | 0.049 | 5.068 | 0.138 | 0.732 |
| | 96 | **0.173** | **0.008** | **0.021** | 0.885 | 0.019 | 0.076 | 1.401 | 0.118 | 0.053 | 8.971 | 0.120 | 0.495 |
| | 192 | **0.295** | **0.008** | **0.026** | 2.440 | 0.196 | 0.111 | 2.509 | 0.116 | 0.071 | 10.70 | 0.713 | 0.907 |
| | 336 | **0.352** | **0.008** | **0.040** | 3.444 | 0.019 | 0.107 | 4.945 | 0.113 | 0.049 | 13.30 | 0.135 | 1.413 |
| Stocks[b] | 24 | 0.378 | 0.029 | 0.082 | **0.313** | **0.012** | 0.072 | 0.341 | 0.034 | 0.119 | 0.314 | 0.012 | **0.071** |
| | 64 | **0.213** | **0.006** | **0.017** | 0.555 | 0.006 | 0.021 | 0.663 | 0.034 | 0.089 | 0.566 | 0.013 | 0.040 |
| | 96 | **0.222** | **0.002** | **0.010** | 0.723 | 0.005 | 0.025 | 1.056 | 0.034 | 0.082 | 0.731 | 0.008 | 0.016 |
| | 192 | **0.285** | **0.003** | **0.007** | 0.899 | 0.004 | 0.024 | 1.282 | 0.033 | 0.065 | 5.057 | 0.026 | 0.150 |
| | 336 | **0.630** | **0.006** | **0.011** | 0.850 | 0.006 | 0.019 | 0.885 | 0.033 | 0.063 | 5.186 | 0.036 | 0.394 |
| fMRI | 24 | **0.472** | 0.022 | 5.122 | 0.496 | 0.036 | **2.613** | 5.851 | 0.152 | 23.38 | 0.547 | 0.117 | 31.72 |
| | 64 | **0.674** | 0.074 | 5.300 | 0.796 | 0.043 | **3.722** | 5.671 | 0.068 | 14.04 | 1.650 | 0.152 | 52.81 |
| | 96 | **1.027** | 0.075 | 4.936 | 1.098 | 0.043 | **3.675** | 5.327 | 0.065 | 11.54 | 2.213 | 0.214 | 49.58 |
| | 192 | **2.919** | 0.077 | 5.262 | 3.649 | 0.044 | **3.670** | 5.120 | 0.088 | 7.761 | 8.363 | 0.379 | 73.81 |
| | 336 | **6.648** | 0.148 | **3.230** | 9.670 | 0.039 | 3.860 | / | / | / | 21.42 | 0.773 | 116.6 |
| Electricity | 24 | 0.046 | 0.030 | **0.505** | **0.035** | **0.015** | 0.640 | 0.176 | 0.031 | 1.745 | 5.103 | 0.112 | 5.936 |
| | 64 | **0.170** | 0.029 | **0.636** | 0.188 | **0.019** | 1.079 | 0.212 | 0.018 | 1.117 | 7.625 | 0.422 | 11.31 |
| | 96 | **0.129** | 0.030 | **0.627** | 0.268 | 0.029 | 1.021 | 0.318 | **0.020** | 0.984 | 18.20 | 0.241 | 34.36 |
| | 192 | **0.064** | 0.022 | **0.461** | 0.426 | 0.051 | 1.241 | 0.972 | 0.027 | 1.359 | 15.92 | 0.120 | 11.22 |
| | 336 | **0.065** | 0.020 | **0.449** | 1.914 | 0.152 | 2.694 | / | / | / | 36.91 | 0.171 | 15.51 |

[a] TimeVAE fails to train on the 336-length fMRI and Electricity datasets because of the prohibitive model size required.

[b] For the Stocks dataset, FDEDiffselects only the 0-frequency component as the low-frequency part.

Figure 2a and Figure 2c show longer windows from the Stocks and the Electricity datasets, respectively. The Stocks data show no periodicity but have significant trends. To handle this, FDEDiffselects only the 0-frequency component as the low-frequency part for a window, *i.e.*, the mean of the time series. The low-frequency model inherently generates the mean values, guiding the high-frequency model to learn the trend information within the Stocks windows. Experiments confirm that such parameter settings effectively handle non-periodic yet highly trending time series, demonstrating the versatility of the FDEDiffframework. Diffusion-TS and TimeVAE also produce high-quality data on the Stocks dataset because they explicitly model polynomial trend components in time series. The 192-length Electricity window in Figure 2c has strong periodicity, enabling FDEDifffirst to generate periodic low-frequency data. Subsequently, under the guidance of the low-frequency part and the enhancement of high-frequency components, FDEDiffgenerates significantly higher-quality data than baseline methods. On the fMRI dataset, all methods struggle to achieve ideal performance (low metric values), primarily due to the high dimensionality that complicates capturing multi-dimensional temporal dependencies.

TimeGAN uses RNNs to encode time series windows and their corresponding temporal latent sequences. However, RNNs suffer from gradient vanishing and long hidden-state propagation paths, making capturing long-term dependencies in longer sequences difficult. Thus, the generation quality of TimeGAN decreases substantially as the window length increases.

## 5.3 ABLATION STUDY

We conduct ablation studies on the ETTh dataset (with window lengths of 96 and 192) to evaluate how the frequency-domain decomposition and enhancement mechanism contributes to the data-generation quality of FDEDiff. There are two critical parameters within this mechanism. The first is the low-frequency ratio, $\alpha$, indicating that the first $\lfloor \alpha W \rfloor$ frequency components are selected as the low-frequency part for a time series of length $W$. The second parameter is the high-frequency enhancement factor, $\lambda$, specifying the degree to which high-frequency components are amplified.

When $\alpha = 1$, all frequency components are treated as low-frequency, meaning the high-frequency model is disabled and the whole frequency-domain mechanism is not applied. $\lambda = 1$ implies no amplification of high-frequency components, indicating that FDEDiffonly applies frequency-domain decomposition without enhancement. When evaluating the impact of varying $\alpha$, we fix $\lambda = 2$; likewise, when examining the effect of different $\lambda$, we fix $\alpha = 0.02$.

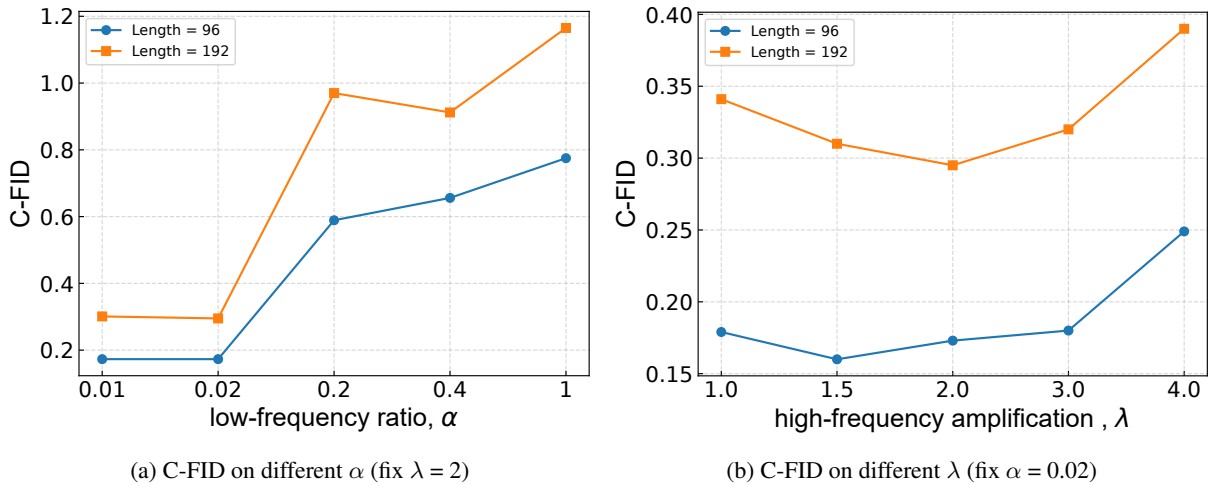

(a) C-FID on different $\alpha$ (fix $\lambda = 2$)      (b) C-FID on different $\lambda$ (fix $\alpha = 0.02$)

Figure 3: Results of ablation study.

Figure 3 reports C-FID scores for FDEDiffunder different parameter settings. Results show that under both window lengths, the model applying frequency-domain decomposition and enhancement with properly selected parameters, *e.g.*, $\alpha = 0.02$ and $\lambda = 2$, outperforms instances where the mechanism is disabled ($\alpha = 1$) or partially effective ($\lambda = 1$). Notably, excessively small or large values for these parameters hinder optimal performance. For example, when $\lambda$ reaches 4, there is a significant increase in C-FID scores, indicating that overly amplifying high-frequency components can suppress learning of low-frequency characteristics, thus deteriorating generation quality.

# 6  CONCLUSION

Time series data serves as a fundamental resource across diverse domains. However, in many real-world scenarios, the acquisition of adequate, large-scale datasets is obstructed by privacy requirements or inherent data scarcity. This paper presents a novel framework that leverages diffusion-based techniques for generating high-fidelity synthetic time series data. Our core innovation lies in explicitly decomposing each sequence into low-frequency and high-frequency domains. By first modeling slow-varying, coarse trends, the model ensures that fundamental seasonal or periodic behaviors are accurately reproduced. We then inject a customized enhancement stage for the high-frequency components, allowing the model to preserve local, fine-grained variations that might otherwise be smoothed away in diffusion-based processes. This two-stage approach counters the typical pitfalls of conventional VAE-based or GAN-based time series generators, namely the challenge of retaining crucial temporal details over extended horizons.

Our extensive experiments on four public datasets corroborate the efficacy and robustness of our approach. Not only does our approach achieve lower C-FID, ACD, and CCD values compared to baseline methods, but it also maintains competitive performance across varying segment lengths and dimensionalities. These qualities are vital in applications featuring complex temporal dependencies, including long-range correlations and abrupt trend shifts. Furthermore, ablation studies validate that both the frequency-domain decomposition and the high-frequency enhancement factor jointly drive performance gains. Striking the right balance in parameter configurations is essential; while the dual-stage design is critical, excessively large or small parameter values introduce their own set of trade-offs.

By offering a more comprehensive solution to time series generation and emphasizing both local and global structures, our approach expands the capabilities of diffusion-based models. We anticipate that this method will be particularly beneficial for scenarios requiring realistic, privacy-conscious data, creating new opportunities for data-driven research and development across multiple sectors.

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

# A APPENDIX / SUPPLEMENTAL MATERIAL

## A.1 C-FID RESULTS WITH ERROR BARS

Table 3: C-FID on Multiple Datasets and Window Lengths.

| Dataset | Length | C-FID ↓ | | | |
|---------|--------|---------|---------|---------|---------|
| | | FDEDiff | Diffusion-TS | TimeVAE | TimeGAN |
| ETTh | 24 | **0.085**±**.002** | 0.138±.006 | 1.697±.162 | 0.799±.066 |
| | 64 | **0.131**±**.008** | 0.248±.013 | 1.124±.101 | 5.068±.636 |
| | 96 | **0.173**±**.013** | 0.885±.082 | 1.401±.123 | 8.971±.859 |
| | 192 | **0.295**±**.013** | 2.440±.157 | 2.509±.168 | 10.70±.710 |
| | 336 | **0.352**±**.028** | 3.444±.246 | 4.945±.311 | 13.30±1.09 |
| Stocks | 24 | 0.378±.126 | **0.313**±**.028** | 0.341±.069 | 0.314±.051 |
| | 64 | **0.213**±**.047** | 0.555±.077 | 0.663±.135 | 0.566±.117 |
| | 96 | **0.222**±**.033** | 0.723±.094 | 1.056±.203 | 0.731±.089 |
| | 192 | **0.285**±**.096** | 0.899±.157 | 1.282±.273 | 5.057±.568 |
| | 336 | **0.630**±**.104** | 0.850±.101 | 0.885±.112 | 5.186±.588 |
| fMRI | 24 | **0.472**±**.050** | 0.496±.006 | 5.851±.412 | 0.547±.029 |
| | 64 | **0.674**±**.026** | 0.796±.006 | 5.671±.109 | 1.650±.813 |
| | 96 | **1.027**±**.028** | 1.098±.048 | 5.327±.455 | 2.213±.454 |
| | 192 | **2.919**±**.133** | 3.649±.188 | 5.120±.713 | 8.363±1.53 |
| | 336 | **6.648**±**.452** | 9.670±.356 | / | 21.42±2.41 |
| Electricity | 24 | 0.046±.008 | **0.035**±**.007** | 0.176±.022 | 5.103±.211 |
| | 64 | **0.170**±**.009** | 0.188±.047 | 0.212±.016 | 7.625±.313 |
| | 96 | **0.129**±**.011** | 0.268±.046 | 0.318±.017 | 18.20±1.13 |
| | 192 | **0.064**±**.012** | 0.426±.048 | 0.972±.170 | 15.92±1.36 |
| | 336 | **0.065**±**.018** | 1.914±.559 | / | 36.91±2.81 |

