# OpenReview forum: "Frequency Decomposition and Enhancement for Time Series Generation Using Diffusion Models"
_ICLR.cc/2026/Conference — Submitted to ICLR 2026_

### Official Review · Reviewer_yeJK · 2025-10-28

**Soundness:** 2
**Presentation:** 3
**Contribution:** 2
**Rating:** 4
**Confidence:** 4

**Summary:**

This paper proposes FDEDiff, a two-stage time series diffusion model that learns low- and high-frequency signals separately. In order to achieve such a capability, FDEDiff first learns a low-frequency model that tries to generate YL. Then, combine YL with high-frequency generation to produce the results Y. Empirical results demonstrated strong performances compared to previous methods.

**Strengths:**

1. This paper explores the two-stage methods, which are less commonly seen in the time series generation problem.
2. The paper is well written, easy to follow.
3. The experiment section compared multiple length choices to demonstrate the strong performance of FDEDiff.

**Weaknesses:**

1. The experiment sections seem to be lightweight, where some of the most challenging datasets are commonly used (energy).
2. The choice of baseline is relatively simple, with only 3 models. Some of the newer and similar models could also be included.
3. The choice of evaluation metrics seems to deviate from those commonly used, like TimeGAN, TimeVAE, and Diffusion TS. It will be beneficiation to include them for better interpretation of the generation quality

**Questions:**

See weaknesses.

---

### Official Review · Reviewer_rHJh · 2025-10-28

**Soundness:** 1
**Presentation:** 1
**Contribution:** 1
**Rating:** 2
**Confidence:** 5

**Summary:**

This paper proposes FDEDiff, a two-stage diffusion framework for time series generation. It decomposes sequences into low and high frequencies using FFT, trains an unconditional diffusion model on low-frequency signals, then uses a conditional diffusion model to generate high-frequency-amplified sequences. Experiments on four datasets show some improvements over baselines on C-FID, ACD, and CCD metrics. The work has limited novelty (combines existing techniques without significant innovation). I appreciate the honesty of the authors by saying the practical limitations of the model (fixed-length generation only, fails on high dimensions, no conditioning).

**Strengths:**

1. The motivation of this paper is clear. It identifies real issues with diffusion models losing high-frequency details during the noise process. This is a problem worth solving.
2. Tests on four diverse datasets on different sequence lengths and compared with both diffusion and GAN-based recent models.
3. Ablation study is present.

**Weaknesses:**

1. The novelty of this work is limited which combines existing techniques without significant innovations. Frequency decomposition for time-series is well-established (FFT, wavelet decomposition), and FIDE [1] already uses frequency enhancement. The paper doesn't clearly explain what's fundamentally new here beyond engineering existing components together.
2. As the author mentioned in the limitation sections of the paper, this work does not support arbitrary length generation, performs poorly on high-dimensional generation tasks and does not support conditional generation.
3. The author mentioned privacy in the abstract and conclusion, however the model has not been evaluated on the basis of privacy. Also no analysis is provided to check if generated samples are authentic versus memorized copies from training data.
4. Table 1 shows FDEDiff uses 128 hidden size with 8 heads while Diffusion-TS uses 96 hidden with 4 heads. This means different parameter counts and computational costs. The paper claims these are "aligned for fairness" but provides no actual parameter count, or training time comparisons. The two-stage training and inference likely adds significant overhead that's completely ignored in the evaluation.
5. The authors should have used Discriminative Score and Predictive Score like presented in the TimeGAN and Diffusion-TS to evaluate the generated samples.
6. Table 2 shows no error bars.
7. Some writing issues - "FDEDiffto" without space, this appeared often in the paper.

#### References
[1] Galib, Asadullah Hill, Pang-Ning Tan, and Lifeng Luo. "Fide: Frequency-inflated conditional diffusion model for extreme-aware time series generation." Advances in Neural Information Processing Systems 37 (2024): 114434-114457.

**Questions:**

1. What are the actual parameter counts for FDEDiff vs Diffusion-TS? What's the training time and inference speed comparison?
2. How should users set $\alpha$ and $\lambda$ for new datasets?
3. Have you checked for memorization? How do you verify generated samples aren't just copies of training data?

---

### Official Review · Reviewer_z1Fx · 2025-10-30

**Soundness:** 2
**Presentation:** 3
**Contribution:** 1
**Rating:** 2
**Confidence:** 4

**Summary:**

This paper introduces FDEDiff (Frequency-Decomposed and Enhanced Diffusion), a diffusion-based framework for time-series generation. The method leverages a two-stage frequency-domain decomposition and filtering strategy to integrate regular temporal dynamics with component-level frequency information. This design aims to synthesize more accurate and structurally consistent time-series signals by explicitly modeling low- and high-frequency components separately.

**Strengths:**

- The paper addresses a significant limitation of existing diffusion-based time-series models - the loss of crucial high-frequency components - by implementing an explicit frequency decomposition strategy.

- The proposed framework is clearly structured and intuitively motivated, which enhances interpretability by separating the modeling of low- and high-frequency components.

- Experimental results demonstrate modest gains on a few public benchmarks (e.g., ETTh, Stocks) in terms of C-FID and correlation metrics, suggesting partial effectiveness of the core decomposition idea.

**Weaknesses:**

- The overall novelty is marginally incremental, primarily combining existing concepts from models like FIDE and Diffusion-TS into a sequential two-stage pipeline rather than presenting a fundamentally new generative or theoretical paradigm.

- The method relies on heuristic, fixed parameters ($\alpha$ and $\lambda$) to define the frequency split boundary and the degree of spectral enhancement, respectively. Since these parameters are critical to the model's behavior, the absence of an adaptive, learned, or theoretically grounded selection mechanism severely limits the robustness and generalizability of the proposed approach across diverse time series.

- The experimental design is limited to only four datasets, critically omitting key domains such as climate, energy, and traffic, which are standard for evaluating frequency-aware or extreme-event modeling.

- The chosen metrics (C-FID, ACD, CCD) focus exclusively on signal smoothness and correlation, completely lacking tail-sensitive metrics (essential for high-frequency/extreme events) or predictive evaluation (critical for utility).

- The absence of a direct comparison with FIDE, a closely related, published frequency-based diffusion model, significantly undermines the fairness and objective interpretability of the reported performance gains.

- The approach fails to scale to complex tasks such as high-dimensional multivariate or conditional generation, a limitation acknowledged by the authors. The lack of explicit modeling for multivariate dependencies significantly constrains the model’s generality and practical utility in real-world scenarios.

**Questions:**

Please refer to the Weaknesses part above.

---

### Official Review · Reviewer_QWsj · 2025-11-01

**Soundness:** 2
**Presentation:** 2
**Contribution:** 2
**Rating:** 2
**Confidence:** 4

**Summary:**

This paper proposes FDEDiff, a two-stage diffusion-based framework for unconditional time series generation.
The model addresses the issue that conventional diffusion models often produce over-smoothed signals and lose high-frequency details due to progressive noise accumulation. FDEDiff decomposes each time series into low-frequency and high-frequency components using a Fourier transform.

**Strengths:**

- Clear and well-motivated **two-stage diffusion design**, tackling a known weakness of diffusion models.
- The paper is **well-written, structured**, and easy to follow; methods are mathematically sound and derivations are correct.
- **Ablation analysis** effectively validates how α and λ influence performance, showing clear trends.

**Weaknesses:**

- [1] **Lack of novelty**
  - The main idea (frequency decomposition followed by hierarchical diffusion) is **conceptually similar to mr-Diff (Shen et al., 2024)** and **FIDE (Galib et al., 2024)**, both of which handle coarse-to-fine or frequency-aware generation.
  - FDEDiff mainly integrates these two ideas, resulting in an **incremental but well-executed extension**, rather than a fundamentally new generative concept.
- [2] **Lack of analysis on why it works well**
  - The paper attributes improvements to “frequency decomposition” and “high-frequency enhancement” but does not **analyze what the model learns** in each frequency band or how this improves sample realism.
- [3] **Performance heavily depends on window size and dataset periodicity**
  - FDEDiff underperforms or matches baselines on short windows (e.g., 24-length Electricity, Stocks) where periodic patterns are absent. The method’s strength appears **specific to periodic or structured time series**, raising concerns about generalization to irregular, non-repetitive domains.
- [4] **No multi-variate dependency modeling**
  - Section 4.5 admits that FDEDiff “lacks explicit modeling for dependencies among multiple variables.” As a result, it performs poorly on high-dimensional datasets like **fMRI (50 dims)** or **Electricity (30 PCA-reduced dims)**, suggesting **limited scalability** to complex multivariate dynamics.
- [5] **Lack of datasets used for the experiments**

**Questions:**

See the weaknesses

---

### Meta-Review · Area_Chair_NBnY · 2026-01-02

**Summary:**

This paper proposes a multi-stage diffusion-based framework for unconditional time series generation, motivated by explicitly modeling low- and high-frequency components. While the experimental results show some improvements over baselines, the reviewers raise concerns regarding the limited novelty of the approach. The core ideas appear closely related to existing techniques, and the paper does not provide new theoretical insights or a clear explanation of why the proposed decomposition and multi-stage methodology should lead to improved generation quality.

The authors did not submit a rebuttal to address these concerns. Given the absence of a response, and the overall evaluation scores, I recommend rejecting the paper.

**Reviewer Concerns:**

No rebuttal was provided.

**Reviewer Scores:**

The main criticism from the reviewers is the limited novelty, and it seems unlikely that the discussion phase would have significantly changed the evaluations.

---

### Decision · Program_Chairs · 2026-01-26

Reject